# High infectiousness immediately before COVID-19 symptom onset highlights the importance of continued contact tracing

William S Hart[1]*, Philip K Maini[1], Robin N Thompson[2,3]

[1]Mathematical Institute, University of Oxford, Oxford, United Kingdom; [2]Mathematics Institute, University of Warwick, Coventry, United Kingdom; [3]Zeeman Institute for Systems Biology and Infectious Disease Epidemiology Research, University of Warwick, Coventry, United Kingdom

## Abstract

**Background:** Understanding changes in infectiousness during SARS-COV-2 infections is critical to assess the effectiveness of public health measures such as contact tracing.

**Methods:** Here, we develop a novel mechanistic approach to infer the infectiousness profile of SARS-COV-2-infected individuals using data from known infector–infectee pairs. We compare estimates of key epidemiological quantities generated using our mechanistic method with analogous estimates generated using previous approaches.

**Results:** The mechanistic method provides an improved fit to data from SARS-CoV-2 infector–infectee pairs compared to commonly used approaches. Our best-fitting model indicates a high proportion of presymptomatic transmissions, with many transmissions occurring shortly before the infector develops symptoms.

**Conclusions:** High infectiousness immediately prior to symptom onset highlights the importance of continued contact tracing until effective vaccines have been distributed widely, even if contacts from a short time window before symptom onset alone are traced.

**Funding:** Engineering and Physical Sciences Research Council (EPSRC).

*For correspondence:
william.hart@keble.ox.ac.uk

**Competing interests:** The authors declare that no competing interests exist.

## Introduction

The precise proportion of SARS-CoV-2 transmissions arising from non-symptomatic (either presymptomatic or asymptomatic) infectors, as well as from unreported infected hosts with only mild symptoms, remains uncertain (*Buitrago-Garcia et al., 2020*; *Casey et al., 2020*). Statistical models can be used to assess the relative contributions of presymptomatic and symptomatic transmission using data from infector–infectee transmission pairs (*Ferretti et al., 2020a*; *Ferretti et al., 2020b*; *Zhang, 2020*; *Liu et al., 2020*; *Tindale et al., 2020*). The distributions of three important epidemiological time periods – the generation time (the difference between the infection times of the infector and infectee) (*Ferretti et al., 2020a*; *Ferretti et al., 2020b*; *Deng et al., 2020*; *Ganyani et al., 2020*), the time from onset of symptoms to transmission (TOST) (*Ferretti et al., 2020b*; *He et al., 2020*; *Ashcroft et al., 2020*), and the serial interval (the difference between the symptom onset times of the infector and infectee) (*Ferretti et al., 2020b*; *Du et al., 2020*) – can also be inferred (*Figure 1A*). The generation time and TOST distributions indicate the average infectiousness of a host at each time since infection and time since symptom onset, respectively (*He et al., 2020*; *Fraser, 2007*). These distributions are important for assessing the effectiveness of public health measures such as isolation (*Ashcroft et al., 2021*; *Wells et al., 2021*) and contact tracing (*Ferretti et al., 2020a*; *Fraser et al., 2004*; *Davis et al., 2020*). Estimates of the SARS-CoV-2 generation time have typically involved an assumption that a host's infectiousness is independent of their

**eLife digest** The risk of a person with COVID-19 spreading the SARS-CoV-2 virus that causes it to others varies over the course of their infection. Transmission depends both on how much virus is in the infected person's airway and their behaviors, such as whether they wear a mask and how many people they have contact with. Learning more about when people are most infectious would help public health officials stop the spread of the virus. For example, officials can then introduce policies that ensure that people are isolated when they are most infectious.

The majority of studies assessing when people with COVID-19 are most infectious so far have assumed that transmission is not linked to when symptoms appear. But that may not be true. After people develop symptoms, they may be more likely to stay home, avoid others, or take other measures that prevent transmission.

Using computer modeling and data from previous studies of individuals who infected others with SARS-CoV-2, Hart et al. show that about 65% of virus transmission occurs before symptoms develop. In fact, the computational experiments show the risk of transmission is highest immediately before symptoms develop. This highlights the importance of identifying people exposed to someone infected with the virus and isolating potential recipients before they develop symptoms.

This information may help public health officials develop more effective strategies to prevent the spread of SARS-CoV-2. It may also help scientists develop more accurate models to predict the spread of the virus. However, the computational experiments used data on infections early in the pandemic that may not reflect the current situation. Changes in public health policy, the behavior of individuals and the appearance of new strains of SARS-CoV-2, all affect the timing of transmission. As more recent data become available, Hart et al. plan to explore how characteristics of transmission have changed as the pandemic has progressed.

symptom status (*Ferretti et al., 2020a*; *Deng et al., 2020*; *Ganyani et al., 2020*; *Knight and Mishra, 2020*; *Lehtinen et al., 2021*; *Figure 1B*, left). However, such an assumption is unjustified (*Lehtinen et al., 2021*; *Bacallado et al., 2020*) and can lead to a poor fit to data (*Ferretti et al., 2020b*).

Here, we develop a mechanistic approach for inferring key epidemiological time periods using data from infector–infectee pairs (*Figure 1B*, right). This approach was motivated by compartmental epidemic models with Gamma distributed stage durations (*Lloyd, 2009*; *Wearing et al., 2005*) and changes in infectiousness during infection (*Hethcote et al., 1991*; *Christofferson et al., 2014*; *Hart et al., 2019*; *Hart et al., 2020*; *Gatto et al., 2020*; *Aleta et al., 2020*). Our method provides an improved fit to data from SARS-CoV-2 transmission pairs compared to previous approaches, namely, (1) a model assuming that transmission and symptoms are independent (*Ferretti et al., 2020a*; *Deng et al., 2020*; *Ganyani et al., 2020*; *Knight and Mishra, 2020*) and (2) a previous statistical method in which this assumption is relaxed (*Ferretti et al., 2020b*). Under our best-fitting model, the proportion of presymptomatic transmissions is high, with many transmissions occurring in a short time window prior to symptom onset. We consider the implications of these results for contact tracing and isolation strategies.

## Results

We considered four different models of infectiousness (see Materials and methods):

i.   The 'variable infectiousness model'. Our mechanistic approach (*Figure 1B*, right panel, solid line) with the relative infectiousness levels for presymptomatic ($P$) and symptomatic ($I$) infectious hosts estimated from the data.
ii.  The 'constant infectiousness model'. Our mechanistic approach (*Figure 1B*, right panel, dashed line), with identical infectiousness levels for presymptomatic ($P$) and symptomatic ($I$) infectious hosts.
iii. The 'Ferretti model'. The best-fitting statistical model from *Ferretti et al., 2020b*, in which the presymptomatic portion of an individual's infectiousness profile is scaled (horizontally) depending on the duration of their incubation period.

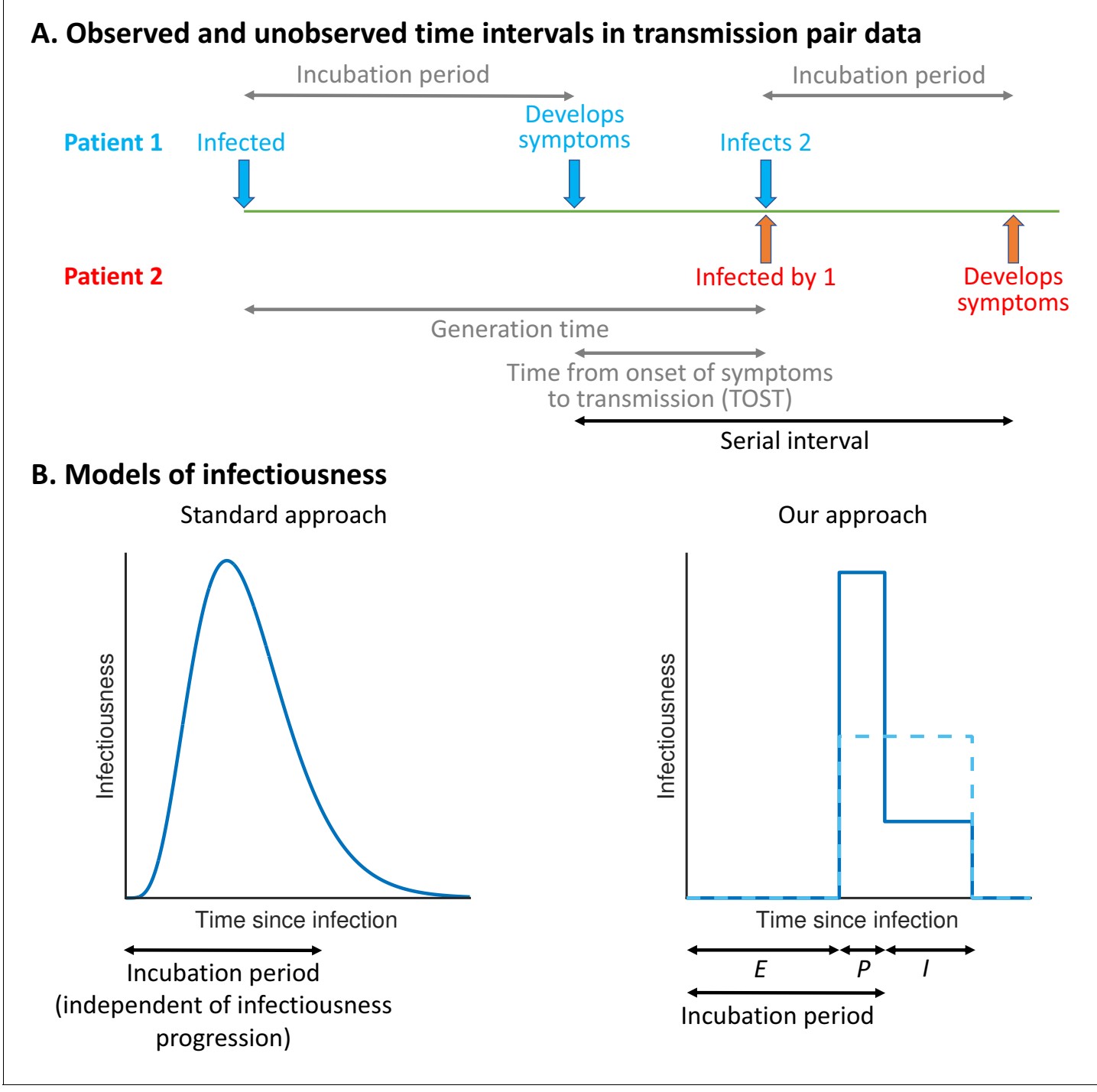

**Figure 1.** Schematic illustrating epidemiological time intervals in data from infector–infectee transmission pairs and approaches for inference from transmission pair data. (A) Transmission pair data generally comprise symptom onset dates for known infector–infectee pairs. These data may be supplemented with partial information about infection times, consisting of a range of possible exposure dates for infectors and/or infectees (*Ferretti et al., 2020a*). While the serial interval for each pair can be calculated directly from the data (with some uncertainty, given the unknown precise times of symptom appearance on the onset dates [*Thompson et al., 2019*]), other time intervals, including the generation time and TOST, are unobserved (these are shown in grey). (B) In standard approaches (left panel) for inferring infectiousness profiles from transmission pair data, the infectiousness of a host at a given time since infection is assumed to be independent of their incubation period. In our approach (right panel), we link a host's infectiousness with when they develop symptoms. We assume that individuals are not infectious during the latent (*E*) period and that infectiousness may either vary between the presymptomatic infectious (*P*) and symptomatic infectious (*I*) periods (solid line – this corresponds to our

*Figure 1 continued on next page*

*Figure 1 continued*

'variable infectiousness model'), for example due to changing behaviour in response to symptoms (*Manfredi and D'Onofrio, 2013*), or be identical in these two time periods (dashed line – this corresponds to our 'constant infectiousness model').

iv. The 'independent transmission and symptoms model'. The standard approach (*Ferretti et al., 2020a*; *Ganyani et al., 2020*; *Figure 1B*, left panel) in which infectiousness is assumed independent of symptoms.

We fitted each model to data from 191 SARS-CoV-2 transmission pairs (*Ferretti et al., 2020b*; *Figure 2—source data 1*) obtained by combining data from five studies (*Ferretti et al., 2020a*; *He et al., 2020*; *Xia et al., 2020*; *Cheng et al., 2020*; *Zhang et al., 2020*). To account for uncertainty in the precise times of symptom appearance within the day of onset for the infector and infectee (*Thompson, 2020*), we used data augmentation Markov chain Monte Carlo (MCMC). Point estimates and credible intervals for model parameters are given in *Supplementary file 1*. The Ferretti model and independent transmission and symptoms model were also fitted to the same data in *Ferretti et al., 2020b* (the parameter estimates obtained in *Ferretti et al., 2020b* lie within the credible intervals shown in *Supplementary file 1*), but estimates of epidemiological quantities obtained using those models were not compared directly in that study.

For each model, we calculated the generation time (*Figure 2A*), TOST (*Figure 2B*), and serial interval (*Figure 2C*) distributions using point estimates for the fitted parameters (*Supplementary file 1*). The empirical serial interval distribution is also plotted in *Figure 2C*, to give an approximate visual indication of the goodness of fit of the different models. However, since the

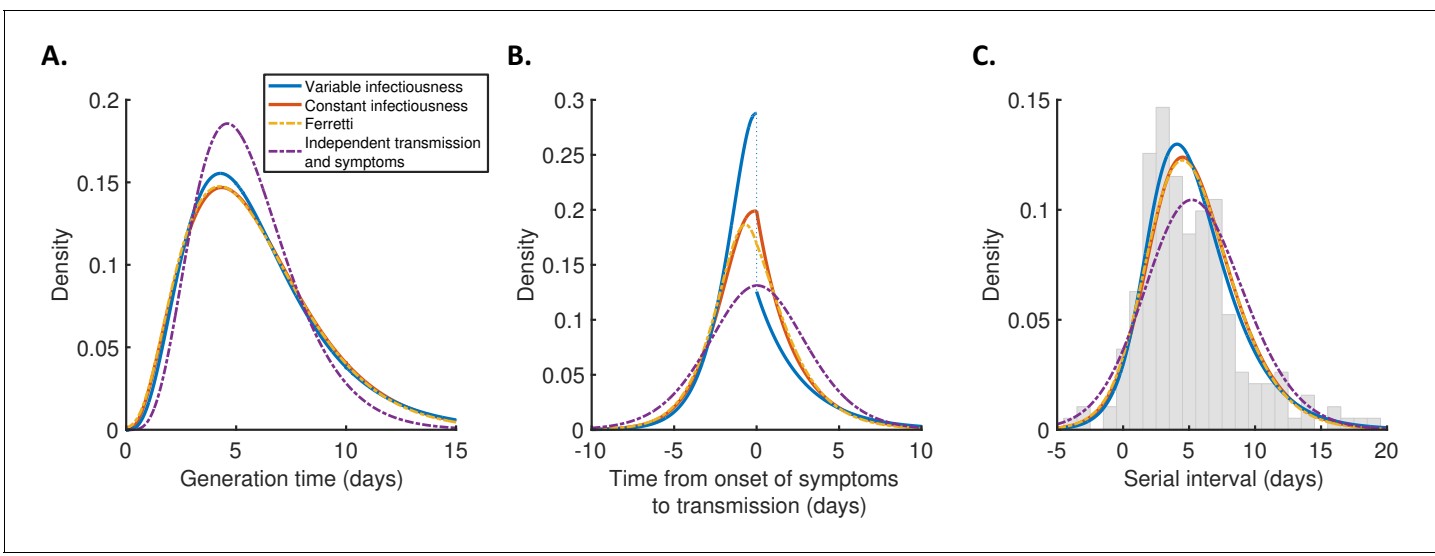

**Figure 2.** Distributions of epidemiological time intervals. Distributions of epidemiological time intervals estimated by fitting different models to data from 191 SARS-CoV-2 transmission pairs (*Figure 2—source data 1*). (A) Generation time, indicating the relative expected infectiousness of a host at each time since infection. (B) Time from onset of symptoms to transmission (TOST), indicating the relative expected infectiousness of a host at each time since symptom onset. (C) Serial interval, indicating the periods between infectors and infectees developing symptoms. In (C), the empirical serial interval distribution from the transmission pair data (*Figure 2—source data 1*) is shown as grey bars. In addition, discretised versions of the serial interval distributions, calculated using the method in *Cori et al., 2013*, are shown in *Figure 2—figure supplement 1*. In all panels, lines represent: variable infectiousness model (blue), constant infectiousness model (red), Ferretti model (orange dashed), and independent transmission and symptoms model (purple dashed). We assumed a specified incubation period distribution (*Lauer et al., 2020*) when fitting the different models to data (see Materials and methods); equivalent panels using an alternative incubation period distribution (*Linton et al., 2020*) are shown in *Figure 2—figure supplement 2*.

The online version of this article includes the following source data and figure supplement(s) for figure 2:

**Source data 1.** Transmission pair data.

**Figure supplement 1.** Discretised serial interval distributions.

**Figure supplement 2.** Robustness to the assumed incubation period distribution.

data contained intervals of possible exposure times in addition to symptom onset dates, this only gives a partial picture of the goodness of fit. Therefore, we also calculated the Akaike information criterion (AIC) for each model. When calculating AIC values, we considered maximum likelihood parameter estimates with symptom onsets occurring in the middle of the onset dates, to avoid comparing models based on likelihoods calculated using augmented data. The best fit to the data was obtained using the variable infectiousness model (ΔAIC = 0). The constant infectiousness model gave the next best fit (ΔAIC = 1.3), followed by the Ferretti model (ΔAIC = 5.1). Finally, the model with the standard assumption of independent transmission and symptoms fitted least well (ΔAIC = 38.9).

The predicted variability in the generation time between individuals was lower for the independent transmission and symptoms model compared to the other three models (*Figure 2A*). On the other hand, the TOST distribution was most concentrated around the time of symptom onset for the best-fitting variable infectiousness model, and least concentrated for the independent transmission and symptoms model (*Figure 2B*). In the best-fitting model, a decrease in infectiousness was inferred following symptom onset, likely due to behavioural factors that reduce the transmission risk following symptom appearance (*Manfredi and D'Onofrio, 2013*).

Using the full posterior distributions of model parameters obtained when fitting the models to data, we calculated posterior estimates of the proportion of transmissions occurring before symptom onset (for hosts who developed symptoms) for each model (*Figure 3A*). The median (95% credible interval) proportion of presymptomatic transmissions was 0.65 (0.53–0.77), 0.56 (0.50–0.62), 0.55 (0.48–0.62), and 0.49 (0.43–0.56) under the variable infectiousness model, constant infectiousness model, Ferretti model, and independent transmission and symptoms model, respectively. The central estimate of 65% of transmissions occurring prior to symptom onset using the best-fitting model is higher than estimated in most previous studies in which the generation time and/or TOST were

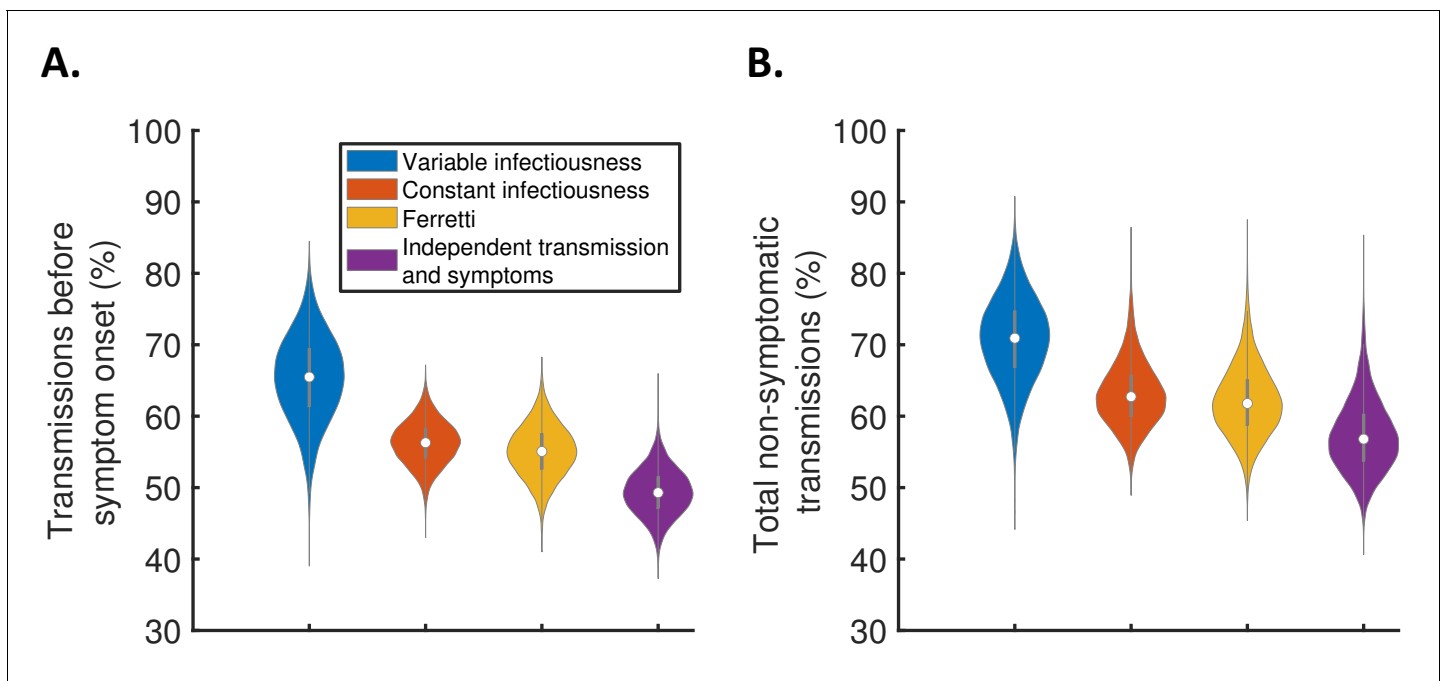

**Figure 3.** The contribution of non-symptomatic infectious individuals to transmission. (**A**) Violin plots indicating posterior distributions for the proportion of transmissions occurring prior to symptom onset for individuals who develop symptoms (i.e., neglecting transmissions from individuals who remain asymptomatic throughout infection) for the different models. (**B**) Posterior distributions for the total proportion of non-symptomatic transmissions, accounting for transmissions from asymptomatic infectious individuals (*Figure 3—figure supplement 1*), for the different models. Equivalent panels assuming an alternative incubation period distribution (*Linton et al., 2020*) are shown in *Figure 3—figure supplement 2*. The online version of this article includes the following figure supplement(s) for figure 3:

**Figure supplement 1.** The contribution of asymptomatic cases to transmission.
**Figure supplement 2.** Robustness to the assumed incubation period distribution.

estimated (*Ferretti et al., 2020a*; *Ferretti et al., 2020b*; *He et al., 2020*; *Ashcroft et al., 2020*). In the wider literature, we note significant variation in estimates of the contribution of presymptomatic transmission (obtained under a range of different modelling assumptions), including estimates exceeding 65% (*Casey et al., 2020*; *Tindale et al., 2020*; *Ganyani et al., 2020*).

We also combined the estimates in *Figure 3A* with the results of a previous study (*Buitrago-Garcia et al., 2020*) in which the extent of asymptomatic transmission (i.e., transmissions from individuals who never display symptoms) was characterised (*Figure 3—figure supplement 1*), to obtain estimates for the total proportion of non-symptomatic (either presymptomatic or asymptomatic) transmissions for the different models (*Figure 3B*). The non-symptomatic proportion was highest for the variable infectiousness model and lowest for the independent transmission and symptoms model.

Finally, we explored the implications of these results for isolation and contact tracing (*Figure 4*), under the simplifying assumptions of perfect isolation (i.e., isolation prevents transmission completely) and perfect contact tracing (i.e., all contacts are traced successfully during periods of contact tracing). Imperfect isolation and contact tracing are considered in *Figure 4—figure supplement 1*. Considering a scenario in which a case (referred to here as the 'index case') is detected following symptom onset, we first calculated how many transmissions from the index case are expected to be prevented for different time delays between the index case developing symptoms and being isolated (*Figure 4A*), compared to a scenario in which the index case is never isolated. We then considered tracing the contacts of that index case, inferring the proportion of presymptomatic contacts identified for different contact elicitation windows (*Figure 4B*). As an example, a contact elicitation window of 2 days means that all contacts of the index case that occurred in the 2 days prior to the index case developing symptoms are traced (in addition to contacts that occurred after the index case developed symptoms). Finally, we considered isolation of infected contacts of

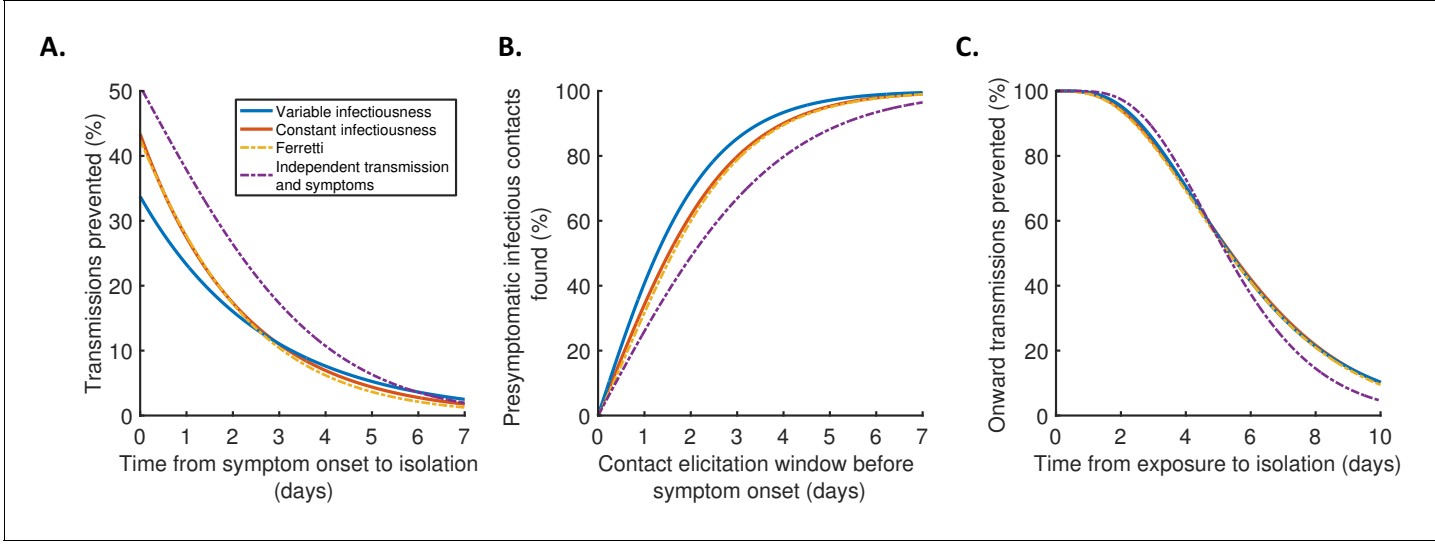

**Figure 4.** Implications for isolation and contact tracing. (A) Effect of the timing of isolation of symptomatic index cases: the proportion of transmissions prevented through isolation, for different time periods between symptom onset and isolation. (B) Effect of the contact elicitation window: the proportion of presymptomatic infectious contacts found for different times up to which contacts are traced before the symptom onset time of the index host. (C) Effect of the timing of isolation of infected contacts: the proportion of onward transmissions generated by the contacts prevented by isolation of those contacts, for different time periods between exposure to the index host and isolation of the contacts. In all panels, lines represent predictions obtained using point estimate parameters for the variable infectiousness model (blue), constant infectiousness model (red), Ferretti model (orange dashed), and independent transmission and symptoms model (purple dashed). Here, isolation and contact tracing are assumed to be 100% effective; equivalent panels in which the effectiveness is less than 100% are shown in *Figure 4—figure supplement 1*. Equivalent panels assuming an alternative incubation period distribution (*Linton et al., 2020*) are shown in *Figure 4—figure supplement 2*.

The online version of this article includes the following figure supplement(s) for figure 4:

**Figure supplement 1.** Robustness to effectiveness of contact tracing and isolation.

**Figure supplement 2.** Robustness to the assumed incubation period distribution.

the index case. We calculated the expected proportion of transmissions generated by those contacts prevented for different time periods between the index case transmitting the virus to the contact and the contact being isolated (*Figure 4C*).

Under the best-fitting variable infectiousness model, 23% (17–31%) of all transmissions that would be generated by a symptomatic host are prevented if the host is isolated one day after symptom onset (*Figure 4A*, blue). This compares to a higher estimate of 38% (32–44%) with the standard independent transmission and symptoms assumption (*Figure 4A*, purple dashed) and intermediate estimates for the constant infectiousness (*Figure 4A*, red) and Ferretti (*Figure 4A*, orange dashed) models. The limited impact of isolation of symptomatic hosts alone under the variable infectiousness model, which is due to the high predicted proportion of presymptomatic transmissions (*Figure 3A*), highlights the need to also conduct contact tracing.

The variable infectiousness model indicates that 69% (57–81%) of presymptomatic infectious contacts are identified if a contact elicitation window of (up to) 2 days before the index host develops symptoms is used (as in the UK [*UK Government, 2021*] and USA [*Centres for Disease Control and Prevention, 2021*]), compared to only 49% (44–53%) for the independent transmission and symptoms model (*Figure 4B*). If the contact elicitation window is extended to 4 days, then 93% (88–97%) of presymptomatic infectious contacts are identified under the variable infectiousness model. However, while choosing a longer contact elicitation window ensures more infected contacts are identified, it also requires more contacts to be traced, many of whom are likely to be uninfected. This effect is enhanced by the fact that index cases are expected to be less infectious at longer time periods prior to symptom onset (*Figure 2B*).

For practical assessments of contact tracing and isolation effectiveness, it may be necessary to consider the combined effects of different delays at each stage of the contact tracing and isolation process. For example, if there is a delay of 2 days between an index case infecting a contact and the index case showing symptoms, and a further delay of 2 days between the index case showing symptoms and the contact being traced and isolated, then this corresponds to a total delay of 4 days between the contact being infected and isolated (assuming that the contact elicitation window is at least 2 days, so that the contact is traced). Under the variable infectiousness model, 71% of onward transmissions from the contact would then be expected to be prevented after this delay (*Figure 4C*). In contrast, for an infectious contact that occurred 4 days before the index host developed symptoms (so that the total delay between the contact being infected and isolated is 6 days, assuming that the contact elicitation window is at least 4 days so the contact is traced), only 41% of the contact's onward infections would be expected to be prevented (*Figure 4C*).

## Discussion

Here, we have considered a range of approaches for estimating epidemiological time periods using data from SARS-CoV-2 infector–infectee transmission pairs. Our mechanistic framework provides an improved fit to data compared to a model predicated on the assumption that infectiousness is independent of symptoms. Despite neglecting potential relationships between viral shedding and symptoms, as well as behavioural changes in response to symptoms (*Manfredi and D'Onofrio, 2013*), that assumption underlies most previous studies in which the SARS-COV-2 generation time distribution has been estimated (*Ferretti et al., 2020a*; *Deng et al., 2020*; *Ganyani et al., 2020*; *Knight and Mishra, 2020*).

Some previous studies in which the generation time (*Ferretti et al., 2020b*; *Davis et al., 2020*) and/or TOST distributions (*Ferretti et al., 2020b*; *He et al., 2020*; *Ashcroft et al., 2020*) were estimated have considered an alternative assumption that infectiousness depends only on the time since symptom onset, independent of the time of infection. If the serial interval is always positive, which is not the case for COVID-19 (*Du et al., 2020*), this is equivalent to assuming that the serial interval and generation time distributions are identical (*Lehtinen et al., 2021*; *Cori et al., 2013*; *Britton and Scalia Tomba, 2019*). In one article (*Ferretti et al., 2020b*), a non-mechanistic model (the Ferretti model) was developed in which a host's infectiousness could depend on both the time since infection and the time since symptom onset. However, as we have demonstrated, our mechanistic approach provides an improved fit to data compared to that model. In addition, our method is useful for parameterising population-scale compartmental epidemic forecasting models, since the time periods derived using our approach correspond naturally to compartments (*Hart et al., 2020*).

It should be noted that an assumption underlying the 'E/P/I' structure of the best-fitting variable infectiousness model (*Figure 1B*, right, solid line) is that infectiousness may change when individuals develop symptoms. The relative infectiousness of presymptomatic and symptomatic infectious individuals is then estimated from the data. Here, we attributed the inferred reduction in transmission following symptom onset found in *Figure 2B* (blue line) to behavioural factors. However, in practice behavioural changes may not occur immediately after symptoms appear, particularly if initial symptoms are mild or non-specific. A delay between symptom onset and a change in infectiousness could in principle be incorporated into our mechanistic framework by adding an additional stage of infection. This would generate a continuous TOST profile. However, we did not take this approach here since such increased model complexity would require additional parameters to be estimated, likely requiring further data.

One caveat of this study is that our estimates were obtained using data collected early in the COVID-19 pandemic (January–March 2020). Since local case numbers were then increasing in locations where some (although not all) of the data were collected (*Ferretti et al., 2020b*), shorter serial intervals may have been over-represented in the dataset (*Britton and Scalia Tomba, 2019*). On the other hand, studies from China have indicated a shortening of the generation time (*Sun et al., 2021*) and serial interval (*Ali et al., 2020*) over time due to non-pharmaceutical interventions, perhaps suggesting longer serial intervals at the beginning of the pandemic. Differences in isolation policies are also likely to affect predictions of the contribution of presymptomatic transmission (*Casey et al., 2020*; *Sun et al., 2021*). We did not explicitly account for isolation policies already in place when the transmission pair data were collected, potentially lowering the estimated effectiveness of isolating symptomatic hosts. More recently, the emergence of novel variants may also have affected the generation time, although their impact is not yet fully clear (*Davies et al., 2021*). Therefore, while our main aim was to compare estimates of key epidemiological quantities under different modelling assumptions, it would be of interest to update our analyses when more recent data from infector–infectee pairs become available.

In summary, using a novel mechanistic approach in combination with data from SARS-CoV-2 infector–infectee pairs to infer key epidemiological quantities indicates that a higher proportion of transmissions occur prior to symptoms than predicted by existing methods. A significant proportion of these transmissions arise immediately before symptom onset. This shows that, while the impact of isolation of symptomatic hosts alone may be limited, combining this with contact tracing and isolation of presymptomatic infected contacts is valuable even if the contact elicitation window is short. The use and refinement of contact tracing programmes in countries worldwide is therefore of clear public health importance.

## Materials and methods

### Notation and general details

Here, we outline the notation used in this section when describing the different models that we considered. For a given transmission pair, we label the infector as 1 and the infectee as 2, and define:

$$
\begin{aligned}
t_{ik} &= (\text{time of infection of host } k), \quad k = 1, 2, \\
t_{sk} &= (\text{time of symptom onset of host } k), \quad k = 1, 2, \\
\tau_{inc,k} &= (\text{incubation period of host } k), \quad k = 1, 2, \\
\tau_{gen} &= (\text{generation time}), \\
x_{tost} &= (\text{time from symptom onset of 1 to transmission to 2 (TOST)}), \\
x_{ser} &= (\text{serial interval}).
\end{aligned}
$$

In the above, $t$ is used to denote calendar times, $\tau$ for time intervals relative to the time of infection, and $x$ for time intervals relative to the time of symptom onset. We denote the probability density functions of the incubation period, generation time, TOST, and serial interval as $f_{inc}, f_{gen}, f_{tost}$, and $f_{ser}$, respectively, and use a capital $F$ for the corresponding cumulative distribution functions.

In addition, we denote the expected infectiousness of a host at time since infection $\tau$ as $\beta(\tau)$, and the expected infectiousness at time since symptom onset $x$ as $b(x)$. These infectiousness profiles are related to the generation time and TOST distributions, respectively, by

$$
\beta(\tau) = \beta_0 f_{gen}(\tau),
$$

$$b(x) = \beta_0 f_{tost}(x).$$

Here, $\beta_0$ corresponds to the expected number of transmissions generated by each host who develops symptoms at some stage during infection, that is, the (instantaneous) reproduction number of such hosts (at least if corrections to the reproduction number within a finite contact network [**Keeling and Grenfell, 2000**; **Enright and Kao, 2018**] can be neglected). However, the exact value of $\beta_0$ has no effect on our analyses, since it simply adds a constant factor to the likelihood function given below. We also let $\beta(\tau \mid \tau_{inc})$ and $b(x \mid \tau_{inc})$ be the expected infectiousness at time $\tau$ since infection and at time $x$ since symptom onset, respectively, conditional on an incubation period of $\tau_{inc}$ (these are related by $\beta(\tau \mid \tau_{inc}) = b(\tau - \tau_{inc} \mid \tau_{inc})$ and $b(x \mid \tau_{inc}) = \beta(x + \tau_{inc} \mid \tau_{inc})$).

We considered several different models for infectiousness (details of individual models are given below). In each model, the conditional infectiousness, $\beta(\tau \mid \tau_{inc})$, or equivalently, $b(x \mid \tau_{inc})$, is specified. The distributions of the generation time and TOST can be recovered from this conditional infectiousness by averaging over the incubation period distribution (which is assumed to be known):

$$\beta(\tau) = \beta_0 f_{gen}(\tau) = \int_0^\infty \beta(\tau \mid \tau_{inc}) f_{inc}(\tau_{inc}) \mathrm{d}\tau_{inc},$$

$$b(x) = \beta_0 f_{tost}(x) = \int_0^\infty b(x \mid \tau_{inc}) f_{inc}(\tau_{inc}) \mathrm{d}\tau_{inc}.$$

Alternative (equivalent) expressions for the generation time and TOST distributions are available for some of the models considered (these are detailed in the "Models of infectiousness" subsection below).

To obtain an expression for the serial interval distribution, we note that

$$x_{ser} = x_{tost} + \tau_{inc,2}.$$

We assume throughout that $x_{tost}$ and $\tau_{inc,2}$ are independent, so that the serial interval distribution is given by the convolution

$$f_{ser}(x_{ser}) = \int_0^\infty f_{tost}(x_{ser} - \tau_{inc}) f_{inc}(\tau_{inc}) \mathrm{d}\tau_{inc}.$$

The proportion of presymptomatic transmissions (out of all transmissions generated by individuals who develop symptoms) can be calculated as

$$q_P = \int_{-\infty}^0 f_{tost}(x_{tost}) \mathrm{d}x_{tost},$$

although simpler equivalent expressions for individual models are also detailed later.

## Data

Following **Ferretti et al., 2020b**, we considered SARS-COV-2 transmission pair data from five different studies (**Ferretti et al., 2020a**; **He et al., 2020**; **Xia et al., 2020**; **Cheng et al., 2020**; **Zhang et al., 2020**), totalling 191 infector–infectee pairs (**Figure 2—source data 1**). In all 191 transmission pairs, both the infector and the infectee developed symptoms, and the symptom onset date of each host was recorded. In four of the five studies (**Ferretti et al., 2020a**; **He et al., 2020**; **Xia et al., 2020**; **Cheng et al., 2020**), intervals of exposure were available for either the infector or infectee (or both), whereas in the other (**Zhang et al., 2020**), only symptom onset dates were recorded.

## Incubation period

In our main analyses, the incubation period was assumed to follow a Gamma distribution with shape parameter 5.807 and scale parameter 0.948 (**Lauer et al., 2020**). This corresponds to a mean incubation period of 5.5 days and a standard deviation of 2.3 days. However, to demonstrate that our main conclusions are robust to the exact incubation period distribution used, we also repeated our analyses using an alternative, more dispersed, Gamma distributed incubation period with a mean of 5.3

days and a standard deviation of 3.2 days (*Linton et al., 2020*; *Figure 2—figure supplement 2*, *Figure 3—figure supplement 2*, *Figure 4—figure supplement 2*).

## Models of infectiousness

### Independent transmission and symptoms model

In this model, the infectiousness of each host at a given time since infection is assumed to be independent of their incubation period, so that

$$\beta(\tau \mid \tau_{inc}) = \beta(\tau) = \beta_0 f_{gen}(\tau),$$

where the generation time distribution, $f_{gen}$, is prescribed. We assumed (*Ferretti et al., 2020a*, *Ganyani et al., 2020*) that

$$\tau_{gen} \sim \text{Gamma}(a, b),$$

where $a$ and $b$ are shape and scale parameters, respectively, so that the mean generation time is $m_{gen} = ab$ and the standard deviation of generation times is $s_{gen} = a^{1/2}b$.

The TOST distribution for this model is given by

$$f_{tost}(x_{tost}) = \int_0^\infty f_{gen}(x_{tost} + \tau_{inc}) f_{inc}(\tau_{inc}) \mathrm{d}\tau_{inc},$$

while the proportion of presymptomatic transmissions is

$$q_P = \int_0^\infty f_{gen}(\tau)(1 - F_{inc}(\tau)) \mathrm{d}\tau.$$

Derivations of these expressions are given in Appendix.

The vector of unknown (log) model parameters, $\theta = \left(\log(m_{gen}), \log(s_{gen})\right)$, was estimated when we fitted the model to the transmission pair data.

### Ferretti model

*Ferretti et al., 2020b* proposed a model in which the conditional infectiousness was specified as the re-scaled skew-logistic distribution,

$$b(x|\tau_{inc}) = \begin{cases} \dfrac{C_F \beta_0 e^{-\left(\frac{xm_{inc}}{\tau_{inc}} - \mu_F\right)/\sigma_F}}{\left(1 + e^{-\left(\frac{xm_{inc}}{\tau_{inc}} - \mu_F\right)/\sigma_F}\right)^{\alpha_F+1}}, & -\tau_{inc} \leq x < 0, \\[3em] \dfrac{C_F \beta_0 e^{-(x-\mu_F)/\sigma_F}}{\left(1 + e^{-(x-\mu_F)/\sigma_F}\right)^{\alpha_F+1}}, & x \geq 0. \end{cases}$$

Here, $m_{inc}$ is the mean incubation period, and $\mu_F$, $\sigma_F$, and $\alpha_F$ are model parameters that do not have straightforward epidemiological interpretations. We set

$$C_F = \frac{\alpha_F}{\sigma_F \left(1 - (1 + e^{(m_{inc}+\mu_F)/\sigma_F})^{-\alpha_F}\right)},$$

in order to ensure the correct scaling for the infectiousness (see Appendix).

The proportion of presymptomatic transmissions is

$$q_P = \frac{\left(1 + e^{\mu_F/\sigma_F}\right)^{-\alpha_F} - \left(1 + e^{(m_{inc}+\mu_F)/\sigma_F}\right)^{-\alpha_F}}{1 - (1 + e^{(m_{inc}+\mu_F)/\sigma_F})^{-\alpha_F}}.$$

A derivation of this expression is given in Appendix.

The vector of unknown model parameters, $\theta = (\mu_F, \log(\sigma_F), \log(\alpha_F))$, was estimated when we fitted the model to the transmission pair data (note that $\mu_F$ could take either positive or negative values, whereas $\sigma_F$ and $\alpha_F$ were constrained to be positive).

## Our mechanistic model

In our mechanistic approach, we divided each infection into three stages: latent ($E$), presymptomatic infectious ($P$), and symptomatic infectious ($I$). The stage durations were assumed to be independent, and infectiousness was assumed to be constant over the duration of each stage. We denote the stage durations by $y_{E/P/I}$, their density and cumulative distribution functions by $f_{E/P/I}$ and $F_{E/P/I}$, and the infectiousness of hosts in the $P$ and $I$ stages by $\beta_{P/I}$, respectively. We also define

$$\alpha = \beta_P/\beta_I$$

to be the ratio of transmission rates in the $P$ and $I$ stages. In this model, the expected number of transmissions generated by each infected host is

$$\beta_0 = \beta_P m_P + \beta_I m_I,$$

where $m_{P/I}$ are the respective mean durations of the $P$ and $I$ stages.

We further assumed that the durations of each stage followed Gamma distributions, with

$$y_E \sim \mathrm{Gamma}\left(k_E, \frac{1}{k_{inc}\gamma}\right),$$

$$y_P \sim \mathrm{Gamma}\left(k_P, \frac{1}{k_{inc}\gamma}\right),$$

$$y_I \sim \mathrm{Gamma}\left(k_I, \frac{1}{k_I\mu}\right),$$

where

$$k_{inc} = k_E + k_P.$$

In particular, the scale parameters of $y_E$ and $y_P$ were both assumed to be equal to $1/(k_{inc}\gamma)$, in order to ensure a Gamma distributed incubation period,

$$\tau_{inc} = y_E + y_P \sim \mathrm{Gamma}\left(k_{inc}, \frac{1}{k_{inc}\gamma}\right).$$

We fixed $k_{inc} = 5.807$ and $\gamma = 1/(5.807 \times 0.948)$, in order to obtain the specified incubation period distribution (see 'Incubation period' subsection above). When we fitted the model to data, we assumed that $k_I = 1$, so that the symptomatic infectious period follows an exponential distribution. The parameters $k_E$ (representing the shape parameter of the latent ($E$) period) and $\mu$ (representing the reciprocal of the mean symptomatic infectious ($I$) period) were estimated in the fitting procedure. We considered two versions of the model: one in which we assumed $\alpha = 1$ (the constant infectiousness model), and one in which $\alpha$ was also estimated (the variable infectiousness model).

For this model, the infectiousness of a host at time $x$ since symptom onset, conditional on an incubation period of $\tau_{inc}$, can be calculated to be

$$b(x|\tau_{inc}) = \begin{cases} \alpha C\beta_0(1 - F_{Beta}(-x/\tau_{inc}; k_P, k_E)), & -\tau_{inc} \leq x < 0, \\ C\beta_0(1 - F_I(x)), & x \geq 0, \end{cases}$$

where $F_{Beta}(s; a, b)$ is the cumulative distribution function of a Beta distributed random variable with shape parameters $a$ and $b$, and

$$C = \frac{\beta_I}{\beta_0} = \frac{k_{inc}\gamma\mu}{\alpha k_P\mu + k_{inc}\gamma}.$$

The TOST distribution is given by

$$f_{tost}(x_{tost}) = \begin{cases} \alpha C(1 - F_P(-x_{tost})), & x_{tost} < 0, \\ C(1 - F_I(x_{tost})), & x_{tost} \geq 0. \end{cases}$$

The generation time can be written as

$$\tau_{gen} = y_E + y^*,$$

where $y^*$ is the time between the start of the $P$ stage and the transmission occurring, and therefore the generation time distribution is given by the convolution

$$f_{gen}(\tau_{gen}) = \int_0^{\tau_{gen}} f^*(\tau_{gen} - y_E) f_E(y_E) \mathrm{d}y_E,$$

where the density, $f^*$, of $y^*$ satisfies

$$f^*(y^*) = C\left( \alpha(1 - F_P(y^*)) + \int_0^{y^*} (1 - F_I(y^* - y_P)) f_P(y_P) \mathrm{d}y_P \right).$$

The proportion of presymptomatic transmissions is

$$q_P = \frac{\beta_P m_P}{\beta_0} = \frac{\alpha k_P \mu}{\alpha k_P \mu + k_{inc}\gamma}.$$

Derivations of these formulae are given in Appendix.

The vector of unknown model parameters, $\theta = (\log(k_E), \log(\mu))$, was estimated when we fitted the constant infectiousness model to the transmission pair data, while the corresponding vector of estimated model parameters for the variable infectiousness model was $\theta = (\log(k_E), \log(\mu), \log(\alpha))$.

## Likelihood and model fitting

For a single transmission pair (labelled $n$), suppose that the times of infection for the infector and infectee are known to lie in the intervals $[t_{i1,L}, t_{i1,R}]$ and $[t_{i2,L}, t_{i2,R}]$, respectively (where these intervals may be infinitely wide), and that their symptom onset times, $t_{s1}$ and $t_{s2}$, are known exactly. In this case (when only that transmission pair is observed), the likelihood of the parameters, $\theta$, of the model of infectiousness under consideration is given by

$$L^{(n)}(\theta) = \frac{1}{\beta_0} \int_{t_{i2,L}}^{t_{i2,R}} \int_{t_{i1,L}}^{t_{i1,R}} b(t_{i2} - t_{s1} \mid t_{s1} - t_{i1}, \theta) f_{inc}(t_{s1} - t_{i1}) f_{inc}(t_{s2} - t_{i2}) \mathrm{d}t_{i1} \mathrm{d}t_{i2},$$

where the dependence of the conditional expected infectiousness, $b(x \mid \tau_{inc}, \theta)$, on the model parameters, $\theta$, is indicated explicitly. A derivation of this expression is given in Appendix. Assuming that each transmission pair in our dataset is independent, the overall likelihood is therefore given by the product of the contributions, $L^{(n)}(\theta)$, from each individual transmission pair, that is,

$$L(\theta) = \prod_{n=1}^{N} L^{(n)}(\theta),$$

where $N$ is the total number of transmission pairs.

To account for uncertainty in the exact symptom onset times within the day of onset (and so avoid imparting bias by fitting continuous-time models to discrete-time symptom onset data), we fitted the models to the data using data augmentation MCMC (*Thompson, 2020*, *Ferguson et al., 2005*, *Cauchemez et al., 2004*). In alternating steps of the chain, we updated either the vector of model parameters, $\theta$, or the exact symptom onset times of each infector and infectee. The chain was run for 2.5 million steps, of which the first 500,000 were discarded as burn-in. Posterior distributions of model parameters were obtained by recording only every 100 iterations of the chain (assuming independent uniform prior distributions for each entry of $\theta$). Point estimates of model parameters (*Supplementary file 1*) were obtained by calculating the posterior mean of $\theta$. Full details of the MCMC procedure are given in Appendix.

In order to provide a straightforward comparison of the goodness of fit between models, we also determined the parameters, $\hat{\theta}$, that maximised the likelihood, $L(\theta)$, for each model under the assumption that each host developed symptoms exactly in the middle of the known onset date. The AIC for each model could then be calculated as

$$\mathrm{AIC} = 2 \times (\text{number of estimated parameters}) - 2\log\left(L\left(\hat{\theta}\right)\right),$$

where three parameters were estimated for the variable infectiousness and Ferretti models, and two parameters for the constant infectiousness and independent transmission and symptoms models. Since the maximum likelihood estimators, $\hat{\theta}$, did not account for uncertainty in exact symptom onset times, they were not used elsewhere in our analyses (however, these all lay within the credible intervals obtained in the MCMC procedure, which are given in *Supplementary file 1*).

## Distributions of the presymptomatic and total non-symptomatic proportion of transmissions

Expressions for the proportion of transmissions, $q_P$, generated prior to symptom onset, are given for the individual models above. Once asymptomatic cases are accounted for, the overall non-symptomatic proportion of transmissions can be written as

$$\frac{p_A x_A + (1 - p_A) q_P}{p_A x_A + (1 - p_A)},$$

where $p_A$ is the proportion of infected individuals who remain asymptomatic and $x_A$ is the ratio between the average number of secondary cases generated by an asymptomatic host and the number generated by a host who develops symptoms at some stage during infection. A derivation of this expression is given in Appendix.

For each model, we used the posterior parameter distributions that were obtained when we fitted the model to data to obtain a sample from the posterior distribution of $q_P$. In order to estimate the total proportion of non-symptomatic transmissions, we assumed the distributions

$$p_A \sim \text{Beta}(85, 186), \qquad [\text{mean } 0.31, \text{ standard deviation } 0.03],$$
$$x_A \sim \text{Lognormal}(-1.04, 0.65^2), \qquad [\text{mean } 0.44, \text{ standard deviation } 0.32],$$

which are consistent with estimates in *Buitrago-Garcia et al., 2020*. These distributions are shown in *Figure 3—figure supplement 1*. We then combined samples from the assumed distributions of $p_A$ and $x_A$ with the sample that we generated from the posterior distribution of $q_P$ to obtain a distribution for the total proportion of non-symptomatic transmissions.

## Contact tracing and isolation

First, we considered the proportion of transmissions that can be prevented if a symptomatic host is isolated $d_1$ days after symptom onset. Assuming that a proportion $\varepsilon_1$ of infectious contacts that would otherwise occur are prevented during the isolation period (and neglecting any transmissions that occur after the end of the isolation period), the overall proportion of transmissions prevented through isolation is

$$\varepsilon_1(1 - F_{tost}(d_1)).$$

We then predicted the proportion of the presymptomatic infectious contacts of a symptomatic index case that will be found, if contacts are traced up to $d_2$ days before the time of symptom onset of the index case. In this scenario, assuming that it is possible to trace a fraction $\varepsilon_2$ of the host's presymptomatic contacts (at times when tracing takes place), then the proportion of presymptomatic infectious contacts found is equal to

$$\frac{\varepsilon_2(q_P - F_{tost}(-d_2))}{q_P}.$$

Finally, we considered the proportion of onward transmissions that can be prevented if an infected individual, who is identified through contact tracing, is isolated $d_3$ days after exposure. Assuming that a proportion $\varepsilon_3$ of infectious contacts that would otherwise occur are prevented during the isolation period, the overall proportion of onward transmissions prevented through isolation is

$$\varepsilon_3(1 - F_{gen}(d_3)).$$

In the main text (*Figure 4*), we assumed that $\varepsilon_1 = \varepsilon_2 = \varepsilon_3 = 1$ (i.e., isolation of symptomatic hosts,

contact identification, and isolation of infected contacts are all 100% effective). Values of $\varepsilon_1$, $\varepsilon_2$, and $\varepsilon_3$ below 1 are considered in *Figure 4—figure supplement 1*.

## Acknowledgements

Thanks to members of the Wolfson Centre for Mathematical Biology at the University of Oxford for useful discussions about this work.

## Additional information

### Funding

| Funder | Grant reference number | Author |
| --- | --- | --- |
| Engineering and Physical Sciences Research Council | Excellence Award | William S Hart |

The funders had no role in study design, data collection and interpretation, or the decision to submit the work for publication.

### Author contributions

William S Hart, Conceptualization, Software, Formal analysis, Investigation, Methodology, Writing - original draft, Writing - review and editing; Philip K Maini, Supervision, Writing - review and editing; Robin N Thompson, Conceptualization, Supervision, Writing - review and editing

### Author ORCIDs

William S Hart ⓘ https://orcid.org/0000-0002-2504-6860
Philip K Maini ⓘ https://orcid.org/0000-0002-0146-9164
Robin N Thompson ⓘ https://orcid.org/0000-0001-8545-5212

### Decision letter and Author response

Decision letter https://doi.org/10.7554/eLife.65534.sa1
Author response https://doi.org/10.7554/eLife.65534.sa2

## Additional files

### Supplementary files

• Supplementary file 1. Table of fitted parameter values. Point estimates (obtained by calculating the posterior mean of the vector of fitted parameters, $\theta$, as described in Materials and methods) and 95% credible intervals for fitted parameters are given for each model. Note that the parameters $\mu_F$ and $\alpha_F$ in the Ferretti model do not have the same epidemiological interpretations as the parameters $\mu$ and $\alpha$ in our mechanistic approach.

• Transparent reporting form

### Data availability

All data generated or analysed during this study are included in the manuscript and supporting files. A source data file has been provided for Figure 2, containing the SARS-CoV-2 transmission pair data used in our analyses. These data were originally reported in references (Ferretti et al., 2020a; He et al., 2020; Xia et al., 2020; Cheng et al., 2020; Zhang et al., 2020), and the combined data were also considered in reference (Ferretti et al., 2020b). Code for reproducing our results is available at https://github.com/will-s-hart/COVID-19-Infectiousness-Profile (copy archived at https://archive.soft-wareheritage.org/swh:1:rev:0e25a4578c650ff22156d18ba899062429cf6ca3).

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

## Appendix 1

### Derivation of the likelihood

For a given transmission pair, the joint probability density that:

i.    patient 1 (the infector) is infected in the time interval $[t_{i1,L}, t_{i1,R}]$;
ii.   patient 1 transmits the pathogen to patient 2 (we write $1 \to 2$ to denote the occurrence of the transmission);
iii.  the transmission from patient 1 to patient 2 occurs in the time interval $[t_{i2,L}, t_{i2,R}]$; and
iv.   patients 1 and 2 develop symptoms at times $t_{s1}$ and $t_{s2}$, respectively;

conditioned on the parameters, $\theta$, of the model of infectiousness under consideration, is given by

$$p\left(1 \to 2, t_{s1}, t_{s2}, [t_{i1,L}, t_{i1,R}], [t_{i2,L}, t_{i2,R}] \mid \theta\right)$$

$$= \int_{t_{i2,L}}^{t_{i2,R}} \int_{t_{i1,L}}^{t_{i1,R}} p(1 \to 2, t_{i1}, t_{s1}, t_{i2}, t_{s2} \mid \theta) \mathrm{d}t_{i1} \mathrm{d}t_{i2}$$

$$= \int_{t_{i2,L}}^{t_{i2,R}} \int_{t_{i1,L}}^{t_{i1,R}} p(1 \to 2, t_{i2}, t_{s2} \mid t_{i1}, t_{s1}, \theta) p(t_{i1}, t_{s1} \mid \theta) \mathrm{d}t_{i1} \mathrm{d}t_{i2}$$

$$= \int_{t_{i2,L}}^{t_{i2,R}} \int_{t_{i1,L}}^{t_{i1,R}} p(t_{s2} \mid 1 \to 2, t_{i1}, t_{s1}, t_{i2}, \theta) p(1 \to 2, t_{i2} \mid t_{i1}, t_{s1}, \theta) p(t_{i1}, t_{s1} \mid \theta) \mathrm{d}t_{i1} \mathrm{d}t_{i2}$$

$$= \int_{t_{i2,L}}^{t_{i2,R}} \int_{t_{i1,L}}^{t_{i1,R}} p(1 \to 2, t_{i2} \mid t_{i1}, t_{s1}, \theta) p(t_{s1} \mid t_{i1}, \theta) p(t_{i1} \mid \theta) p(t_{s2} \mid t_{i2}, \theta) \mathrm{d}t_{i1} \mathrm{d}t_{i2}.$$

We note that

$$p(1 \to 2, t_{i2} \mid t_{i1}, t_{s1}, \theta) \propto b(t_{i2} - t_{s1} \mid t_{s1} - t_{i1}, \theta).$$

This is because the left-hand side gives the probability density of a transmission from 1 to 2 occurring at time $t_{i2}$, conditioned on the infection and onset times of 1, and is therefore proportional to the conditional infectiousness, $b(x_{tost} \mid \tau_{inc}, \theta)$. We also have that

$$p(t_{sk} \mid t_{ik}, \theta) = f_{inc}(t_{sk} - t_{ik}),$$

for $k = 1, 2$. In an exponentially growing epidemic with growth rate $r$, the term $p(t_{i1} \mid \theta)$ will introduce a factor proportional to $e^{rt_{i1}}$ into the likelihood (*Ferretti et al., 2020a*), although we neglect this correction here (note that we found a similar fit to data using the Ferretti model compared to that obtained in *Ferretti et al., 2020b*, in which the same model was fitted to the same dataset with this correction included). We therefore obtain the expression for the likelihood, $L^{(n)}(\theta)$, given in Materials and methods, up to a constant scaling factor. The factor $1/\beta_0$ was added for convenience, although we note that in general,

$$\frac{1}{\beta_0} \int_0^\infty b(x_{tost} \mid \tau_{inc}, \theta) \mathrm{d}x_{tost}$$

may not be equal to 1, since the expected number of secondary infections generated by a host may depend on their incubation period.

### Details of model fitting procedure

We denote the vector of model parameters for the model of infectiousness under consideration by $\theta$, the vectors of symptom onset times for each infector and infectee by $\mathbf{t}_{s1}$ and $\mathbf{t}_{s2}$, and the corresponding likelihood by

$$L(\theta; \boldsymbol{t}_{s1}, \boldsymbol{t}_{s2}) = \prod_{n=1}^{N} L^{(n)}\left(\theta; t_{s1}^{(n)}, t_{s2}^{(n)}\right).$$

In this expression, $L^{(n)}\left(\theta; t_{s1}^{(n)}, t_{s2}^{(n)}\right)$ is the contribution to the likelihood from transmission pair $n$, and $t_{s1}^{(n)}$ and $t_{s2}^{(n)}$ are the symptom onset times of the corresponding infector and infectee (i.e., the $n^{\text{th}}$ entries of $\boldsymbol{t}_{s1}$ and $\boldsymbol{t}_{s2}$, respectively). We define the proposal distributions $Q_1\left(\theta_{prop}|\theta\right)$ and $Q_2^{(n)}\left(t_{s1,prop}^{(n)}, t_{s2,prop}^{(n)}|t_{s1}^{(n)}, t_{s2}^{(n)}\right)$, which are taken to be symmetric (i.e., $Q_1\left(\theta_{prop}|\theta\right) = Q_1\left(\theta|\theta_{prop}\right)$ and $Q_2^{(n)}\left(t_{s1,prop}^{(n)}, t_{s2,prop}^{(n)}|t_{s1}^{(n)}, t_{s2}^{(n)}\right) = Q_2^{(n)}\left(t_{s1}^{(n)}, t_{s2}^{(n)}|t_{s1,prop}^{(n)}, t_{s2,prop}^{(n)}\right)$; the exact proposal distributions we used are detailed below).

The data augmentation MCMC algorithm that we used is given by the following steps:

1. Initialise $\theta = \theta_0$, $\boldsymbol{t}_{s1} = \boldsymbol{t}_{s1,0}$ and $\boldsymbol{t}_{s2} = \boldsymbol{t}_{s2,0}$.
2. For $n = 1, \ldots, N$, calculate $L_0^{(n)} = L^{(n)}\left(\theta_0; t_{s1,0}^{(n)}, t_{s2,0}^{(n)}\right)$.
3. Calculate $L_0 = \prod_{n=1}^{N} L_0^{(n)}$.
4. For $m = 1, \ldots, M$:
   - If $m$ is odd, then:
     - Sample $\theta_{prop}$ from $Q_1\left(\theta_{prop}|\theta_{m-1}\right)$.
     - Set $\boldsymbol{t}_{s1,m} = \boldsymbol{t}_{s1,(m-1)}$ and $\boldsymbol{t}_{s2,m} = \boldsymbol{t}_{s2,(m-1)}$.
     - For $n = 1, \ldots, N$, calculate $L_{prop}^{(n)} = L^{(n)}\left(\theta_{prop}; t_{s1,m}^{(n)}, t_{s2,m}^{(n)}\right)$.
     - Calculate $L_{prop} = \prod_{n=1}^{N} L_{prop}^{(n)}$.
     - Generate a random number, $r$, uniformly distributed between 0 and 1.
     - If $r \leq L_{prop}/L_{m-1}$, set $\theta_m = \theta_{prop}$, $L_m^{(n)} = L_{prop}^{(n)}$ for each $n$, and $L_m = L_{prop}$. Otherwise, set $\theta_m = \theta_{m-1}$, $L_m^{(n)} = L_{m-1}^{(n)}$ for each $n$, and $L_m = L_{m-1}$.
   - If $m$ is even, then:
     - Set $\theta_m = \theta_{m-1}$.
     - For $n = 1, \ldots, N$:
       - Sample $t_{s1,prop}^{(n)}$ and $t_{s2,prop}^{(n)}$ from $Q_2^{(n)}\left(t_{s1,prop}^{(n)}, t_{s2,prop}^{(n)}|t_{s1,(m-1)}^{(n)}, t_{s2,(m-1)}^{(n)}\right)$.
       - Calculate $L_{prop}^{(n)} = L^{(n)}\left(\theta_m; t_{s1,prop}^{(n)}, t_{s2,prop}^{(n)}\right)$.
       - Generate a random number, $r$, uniformly distributed between 0 and 1.
       - If $r \leq L_{prop}^{(n)}/L_{m-1}^{(n)}$, set $t_{s1,m}^{(n)} = t_{s1,prop}^{(n)}$, $t_{s2,m}^{(n)} = t_{s2,prop}^{(n)}$ and $L_m^{(n)} = L_{prop}^{(n)}$. Otherwise, set $t_{s1,m}^{(n)} = t_{s1,(m-1)}^{(n)}$, $t_{s2,m}^{(n)} = t_{s2,(m-1)}^{(n)}$ and $L_m^{(n)} = L_{m-1}^{(n)}$.
     - Calculate $L_m = \prod_{n=1}^{N} L_m^{(n)}$.

We constrained the symptom onset time, $t_s$, of each host to lie on the grid

$$\left[t_{s,L} + \delta t, t_{s,L} + 2\delta t, \ldots, t_{s,L} + 1\right],$$

where $t_{s,L}$ is the start of the day of onset for that host, and we took $\delta t = 0.125$ days. The contribution to the likelihood from each transmission pair, $L^{(n)}\left(\theta; t_{s1}^{(n)}, t_{s2}^{(n)}\right)$, was then calculated by discretising the integrals (see the 'Likelihood and model fitting' subsection in Materials and methods), with the infection time, $t_i$, of a given host constrained to the grid

$$\left[t_{i,L} + \frac{\delta t}{2}, \ldots, t_{i,R} - \frac{\delta t}{2}\right],$$

where $t_{i,L}$ and $t_{i,R}$ are lower/upper bounds for the infection time of that host. Different discretisations were used for the infection and onset times, both to avoid conditioning on an incubation period of zero days (since the conditional infectiousness may be undefined in this case) and to avoid the possibility of transmissions occurring at the exact time of symptom onset (since the infectiousness profile

was allowed to be discontinuous at the onset time in our mechanistic model). We also assumed a maximum possible incubation period of 30 days.

For each model we considered, the initial parameter values, $\theta_0$, were chosen arbitrarily. The initial symptom onset times, $t_{s1,0}$ and $t_{s2,0}$, were uniformly and independently sampled on the grid of possible onset times for each host. Independent normal proposal distributions were used for each entry of $\theta$ – that is, for each individual parameter $\theta^{(j)}$, we set

$$\theta_{prop}^{(j)} = \theta_{current}^{(j)} + r,$$

where $r$ is a normally distributed random variate with mean zero and standard deviation $\sigma^{(j)}$. The tuning parameters, $\sigma^{(j)}$, were chosen to ensure an acceptance rate of between 25% and 30%. We sampled the proposed symptom onset times for each host, $t_{s1,prop}^{(n)}$ and $t_{s2,prop}^{(n)}$, uniformly on the grid of possible onset times for the host under consideration (independently both of the corresponding times in the previous step of the chain, and of the onset times of all other hosts).

## Model-specific derivations

### Independent transmission and symptoms model

For the independent transmission and symptoms model, the TOST distribution is given by

$$
\begin{aligned}
f_{tost}(x_{tost}) &= \frac{1}{\beta_0} \int_0^\infty b(x_{tost} \mid \tau_{inc}) f_{inc}(\tau_{inc}) \mathrm{d}\tau_{inc} \\
&= \frac{1}{\beta_0} \int_0^\infty \beta(x_{tost} + \tau_{inc} \mid \tau_{inc}) f_{inc}(\tau_{inc}) \mathrm{d}\tau_{inc} \\
&= \int_0^\infty f_{gen}(x_{tost} + \tau_{inc}) f_{inc}(\tau_{inc}) \mathrm{d}\tau_{inc}.
\end{aligned}
$$

Alternatively, this formula can be derived by noting that

$$x_{tost} = \tau_{gen} - \tau_{inc,1}.$$

In this model, $\tau_{gen}$ and $\tau_{inc,1}$ are assumed to be independent, so the TOST distribution is therefore given by the convolution of the distributions of $\tau_{gen}$ and $-\tau_{inc,1}$.

The proportion of presymptomatic transmissions is given by

$$
\begin{aligned}
q_P &= \int_{-\infty}^0 f_{tost}(x_{tost}) \, \mathrm{d}x_{tost} \\
&= \int_{-\infty}^0 \int_0^\infty f_{gen}(x_{tost} + \tau_{inc}) f_{inc}(\tau_{inc}) \mathrm{d}\tau_{inc} \mathrm{d}x_{tost} \\
&= \int_0^\infty \int_{\tau_{gen}}^\infty f_{gen}(\tau_{gen}) f_{inc}(\tau_{inc}) \mathrm{d}\tau_{inc} \mathrm{d}\tau_{gen} \\
&= \int_0^\infty f_{gen}(\tau_{gen}) (1 - F_{inc}(\tau_{gen})) \mathrm{d}\tau_{gen}.
\end{aligned}
$$

### Ferretti model

To derive the correct scaling factor, $C_F$, in the conditional infectiousness, we note that we require

$$\int_{-\infty}^\infty f_{tost}(x) \mathrm{d}x = \int_{-\infty}^\infty \int_0^\infty \frac{1}{\beta_0} b(x \mid \tau_{inc}) f_{inc}(\tau_{inc}) \mathrm{d}\tau_{inc} \mathrm{d}x = 1.$$

Now, we can calculate

$$\int_{-\infty}^{\infty} \frac{1}{\beta_0} b(x \mid \tau_{inc}) \mathrm{d}x$$

$$= \int_{-\tau_{inc}}^{0} \frac{C_F e^{-\left(\frac{xm_{inc}}{\tau_{inc}} - \mu_F\right)/\sigma_F}}{\left(1 + e^{-\left(\frac{xm_{inc}}{\tau_{inc}} - \mu_F\right)/\sigma_F}\right)^{\alpha_F + 1}} \mathrm{d}x + \int_{0}^{\infty} \frac{C_F e^{-(x - \mu_F)/\sigma_F}}{\left(1 + e^{-(x - \mu_F)/\sigma_F}\right)^{\alpha_F + 1}} \mathrm{d}x$$

$$= \frac{C_F \sigma_F}{\alpha_F} \left[ 1 - \left(1 + e^{\mu_F/\sigma_F}\right)^{-\alpha_F} + \frac{\tau_{inc}}{m_{inc}} \left( \left(1 + e^{\mu_F/\sigma_F}\right)^{-\alpha_F} - \left(1 + e^{(m_{inc} + \mu_F)/\sigma_F}\right)^{-\alpha_F} \right) \right].$$

Therefore,

$$\int_{-\infty}^{\infty} \int_{0}^{\infty} \frac{1}{\beta_0} b(x \mid \tau_{inc}) f_{inc}(\tau_{inc}) \mathrm{d}\tau_{inc} \mathrm{d}x$$

$$= \int_{0}^{\infty} \left( \int_{-\infty}^{\infty} \frac{1}{\beta_0} b(x \mid \tau_{inc}) \mathrm{d}x \right) f_{inc}(\tau_{inc}) \mathrm{d}\tau_{inc}$$

$$= \frac{C_F \sigma_F}{\alpha_F} \left[ 1 - \left(1 + e^{(m_{inc} + \mu_F)/\sigma_F}\right)^{-\alpha_F} \right] = 1,$$

so we have

$$C_F = \frac{\alpha_F}{\sigma_F \left(1 - \left(1 + e^{(m_{inc} + \mu_F)/\sigma_F}\right)^{-\alpha_F}\right)}.$$

The proportion of presymptomatic transmissions is given by

$$q_P = \int_{-\infty}^{0} f_{tost}(x) \mathrm{d}x$$

$$= \int_{0}^{\infty} \int_{-\infty}^{0} \frac{1}{\beta_0} b(x \mid \tau_{inc}) f_{inc}(\tau_{inc}) \mathrm{d}x \mathrm{d}\tau_{inc}$$

$$= \int_{0}^{\infty} \frac{C_F \sigma_F \tau_{inc}}{\alpha_F m_{inc}} \left[ \left(1 + e^{\mu_F/\sigma_F}\right)^{-\alpha_F} - \left(1 + e^{(m_{inc} + \mu_F)/\sigma_F}\right)^{-\alpha_F} \right] f_{inc}(\tau_{inc}) \mathrm{d}\tau_{inc}$$

$$= \frac{\left(1 + e^{\mu_F/\sigma_F}\right)^{-\alpha_F} - \left(1 + e^{(m_{inc} + \mu_F)/\sigma_F}\right)^{-\alpha_F}}{1 - \left(1 + e^{(m_{inc} + \mu_F)/\sigma_F}\right)^{-\alpha_F}}.$$

## Our mechanistic model

In our mechanistic model, the expected infectiousness of a host at time $x$ since symptom onset is given by

$$b(x) = \begin{cases} \beta_P \times p(Y_P \geq -x), & x < 0, \\ \beta_I \times p(Y_I \geq x), & x \geq 0, \end{cases}$$

where we here explicitly distinguish the random variables $Y_{E/P/I}$ from their observed values $y_{E/P/I}$ (i.e., the lengths of each stage of infection). Therefore,

$$f_{tost}(x_{tost}) = \frac{1}{\beta_0} b(x_{tost}) = \begin{cases} \alpha C (1 - F_P(-x_{tost})), & x_{tost} < 0, \\ C (1 - F_I(x_{tost})), & x_{tost} \geq 0, \end{cases}$$

where

$$C = \frac{\beta_I}{\beta_0} = \frac{\beta_I}{\beta_P m_P + \beta_I m_I} = \frac{1}{\alpha m_P + m_I} = \frac{1}{\left(\frac{\alpha k_P}{k_{inc} \gamma} + \frac{1}{\mu}\right)} = \frac{k_{inc} \gamma \mu}{\alpha k_P \mu + k_{inc} \gamma}.$$

Conditional on an incubation period of length $\tau_{inc}$, the expected infectiousness is

$$b(x \mid \tau_{inc}) = \begin{cases} \beta_P \times p(Y_P \geq -x \mid Y_E + Y_P = \tau_{inc}), & -\tau_{inc} \leq x < 0, \\ \beta_I \times p(Y_I \geq x), & x \geq 0. \end{cases}$$

Now,

$$\begin{aligned} p(Y_P \geq -x | Y_E + Y_P = \tau_{inc}) &= \int_{-x}^{\infty} p(Y_P = y_P | Y_E + Y_P = \tau_{inc}) \mathrm{d}y_P \\ &= \int_{-x}^{\infty} \frac{p(Y_E + Y_P = \tau_{inc} \mid Y_P = y_P) p(Y_P = y_P)}{p(Y_E + Y_P = \tau_{inc})} \mathrm{d}y_P \\ &= \int_{-x}^{\infty} \frac{f_E(\tau_{inc} - y_P) f_P(y_P)}{f_{inc}(\tau_{inc})} \mathrm{d}y_P, \end{aligned}$$

where we used Bayes' rule to obtain the second equality. For the special case of Gamma distributed stage durations considered, we have that

$$\frac{f_E(\tau_{inc} - y_P) f_P(y_P)}{f_{inc}(\tau_{inc})} = \frac{1}{\tau_{inc}} f_{Beta}(y_P / \tau_{inc}; k_P, k_E),$$

where $f_{Beta}(x; a, b)$ is the probability density function of a Beta distributed random variable with shape parameters $a$ and $b$. Therefore,

$$p(Y_P \geq -x | Y_E + Y_P = \tau_{inc}) = F_{Beta}(-x / \tau_{inc}; k_P, k_E),$$

and so

$$b(x | \tau_{inc}) = \begin{cases} \alpha C \beta_0 (1 - F_{Beta}(-x / \tau_{inc}; k_P, k_E)), & -\tau_{inc} \leq x < 0, \\ C \beta_0 (1 - F_I(x)), & x \geq 0. \end{cases}$$

The expected infectiousness at time $y^*$ since the start of the $P$ stage is equal to

$$b^*(y^*) = \beta_P \times p(Y_P \geq y^*) + \beta_I \times p(Y_P \leq y^*, Y_P + Y_I \geq y^*).$$

The second probability can be evaluated by conditioning on the value of $Y_P$, to obtain

$$\begin{aligned} b^*(y^*) &= \beta_P(1 - F_P(y^*)) + \beta_I \int_0^{y^*} p(Y_P \leq y^*, Y_P + Y_I \geq y^* | Y_P = y_p) f_P(y_P) \mathrm{d}y_P \\ &= \beta_P(1 - F_P(y^*)) + \beta_I \int_0^{y^*} p(Y_I \geq y^* - y_P \mid Y_P = y_p) f_P(y_P) \mathrm{d}y_P \\ &= \beta_P(1 - F_P(y^*)) + \beta_I \int_0^{y^*} (1 - F_I(y^* - y_P)) f_P(y_P) \mathrm{d}y_P. \end{aligned}$$

Therefore, the distribution of the time between the start of the $P$ stage and secondary transmission occurring is

$$f^*(y^*) = C \left( \alpha(1 - F_P(y^*)) + \int_0^{y^*} (1 - F_I(y^* - y_P)) f_P(y_P) \mathrm{d}y_P \right).$$

The proportion of presymptomatic transmissions is

$$q_P = \frac{\beta_P m_P}{\beta_0} = \frac{\beta_P m_P}{\beta_P m_P + \beta_I m_I} = \frac{\alpha m_P}{\alpha m_P + m_I} = \frac{\left( \frac{\alpha k_P}{k_{inc} \gamma} \right)}{\left( \frac{\alpha k_P}{k_{inc} \gamma} + \frac{1}{\mu} \right)} = \frac{\alpha k_P \mu}{\alpha k_P \mu + k_{inc} \gamma}.$$

## Total proportion of non-symptomatic transmissions accounting for asymptomatic cases

Here, we derive an expression for the total proportion of non-symptomatic transmissions once asymptomatic cases are accounted for. The (instantaneous) reproduction number, $R$, can be decomposed as

$$R = p_A R_A + (1 - p_A)(R_P + R_I),$$

where $p_A$ is the proportion of completely asymptomatic cases, $R_A$ is the expected number of secondary transmissions generated by each asymptomatic host, and $R_{P/I}$ are the expected numbers of transmissions generated before and after symptom onset by a host who develops symptoms, respectively. The total proportion of non-symptomatic transmissions is given by

$$\frac{p_A R_A + (1-p_A)R_P}{R} = \frac{p_A R_A + (1-p_A)R_P}{p_A R_A + (1-p_A)(R_P + R_I)}$$
$$= \frac{p_A x_A + (1-p_A)q_P}{p_A x_A + (1-p_A)},$$

where

$$q_P = \frac{R_P}{R_P + R_I}$$

is the proportion of transmissions generated prior to symptom onset by hosts who develop symptoms, and

$$x_A = \frac{R_A}{R_P + R_I}$$

is the ratio between the expected number of transmissions generated by an asymptomatic host and the expected number of transmissions generated by a host who develops symptoms.

