## [Decision Letter]

**Acceptance summary:**

The manuscript uses a new approach to model the infectiousness profile of COVID-19 infected individuals. The key finding from this work is that contact tracing prevents a large proportion of onward transmissions, even if contacts within a short window (2 days) are traced. The evidence from this work emphasises the importance of contact tracing and isolation in containing the spread of COVID-19. This finding is of great interest to public health policy makers. The methodology is of general interest to modellers working on COVID-19.

**Decision letter after peer review:**

Thank you for submitting your article "High infectiousness immediately before COVID-19 symptom onset highlights the importance of continued contact tracing" for consideration by *eLife*. Your article has been reviewed by 3 peer reviewers, and the evaluation has been overseen by a Reviewing Editor and a Senior Editor. The following individuals involved in review of your submission have agreed to reveal their identity: Rowland Raymond Kao (Reviewer #2); Elizabeth Lee (Reviewer #3).

As is customary in *eLife*, the reviewers have discussed their critiques with one another. What follows below is the Reviewing Editor's edited compilation of the essential and ancillary points provided by reviewers in their critiques and in their interaction post-review. Please submit a revised version that addresses these concerns directly. Although we expect that you will address these comments in your response letter, we also need to see the corresponding revision in the text of the manuscript. Some of the reviewers' comments may seem to be simple queries or challenges that do not prompt revisions to the text. Please keep in mind, however, that readers may have the same perspective as the reviewers. Therefore, it is essential that you attempt to amend or expand the text to clarify the narrative accordingly.

Essential revisions:

1. Is it possible to use more recent data? If so, the results should be updated. If not, the discussion should be expanded to highlight this caveat, otherwise the conclusion on the effectiveness of contact tracing could be overly optimistic.

2. The model code and compiled data should be made available for public use.

3. The fundamental problem is already well known, and the application to COVID-19, while useful, is better than poorer models, but only marginally better performing than the Ferretti model. The serial interval estimates are only slightly better (figure 2), there are 84% of contacts when considering tracing two days prior to symptoms, compared to what looks like about 80% for the alternative in figure 4 and by the looks of the violin plots from figure 3, quite a bit of overlap if one considers credible intervals. As such, while the analysis is a solid, useful addition to the literature, the authors should provide a better exposition on how it advances scientific insight (the fundamental issues regarding exponential distributions having been identified previously), methodologically (given the thorough analysis by Fraser et al. in 2004) or in terms of impact (given the limited improvement over the Ferretti model).

4. The framing of the paper seems very focused on improving fits to the transmission pair data, however, and I think it would be more impactful to consider the implications of poor estimation of pre-symptomatic transmission and the generation time. I think this shift in focus could also help strengthen the narrative of the paper, which wavers between focusing on model fitting and the importance of implications for contact tracing. Can the authors comment?

5. I was a bit lost in the application of the models to the contact tracing example. The definition of the contact elicitation window (lines 142-144), where identification of contacts would occur up to x days prior to contact symptom onset, makes sense theoretically in this model comparison setting, but it is hard to translate these findings to real-world application. Are there any implications that could be useful for informing contact elicitation strategy (e.g., for how many days after time of infection or symptom onset could contact tracing have a measurable benefit in preventing onward transmissions?)

6. Lines 147-151: Given that the impact on onward transmission events is so dependent on the contact tracing assumptions, I would recommend stating the assumptions explicitly here, reporting the results in relative terms as compared to a single model, or both.

7. How different are the variable infectiousness model results from parameter estimates from the original studies that reported the transmission pairs data?

8. Can the authors comment on the plausibility of the infectiousness distribution in their new proposed models? While better model fitting certainly provides a measurable improvement to leveraging existing data, I'm not aware of studies that support the discontinuous assumptions about infectiousness made here.

9. Assuming alpha means the same thing across the models, why is the 95% credible interval so large for the Feretti model? In general, the model parameters should be more clearly explained for this model. Can the authors comment?

---

## [Author Response]

Essential revisions:1. Is it possible to use more recent data? If so, the results should be updated. If not, the discussion should be expanded to highlight this caveat, otherwise the conclusion on the effectiveness of contact tracing could be overly optimistic.

We thank the reviewer for highlighting the important caveat that the data were collected during the early months of the pandemic. Although further transmission pair data are available from a similar time period, we are not aware of more recent publicly available data. We have therefore discussed this caveat in the revised manuscript as suggested (lines 395-413).

2. The model code and compiled data should be made available for public use.

We thank the reviewer for emphasising the importance of making data and code publicly available. The SARS-CoV-2 transmission pair data underlying our analyses are included in our revised submission as a Source Data file for Figure 2 (the observed serial intervals are represented as a histogram in Figure 2C), and the model code is publicly available at https://github.com/will-s-hart/COVID-19-Infectiousness-Profile/ (this link is provided in the Data Availability section of our revised submission). We have also added a reference to the Source Data file in the Materials and methods of the revised manuscript (line 499), as well as in the caption to Figure 2 (line 952-953) and in the Results (lines 116-117).

3. The fundamental problem is already well known, and the application to COVID-19, while useful, is better than poorer models, but only marginally better performing than the Ferretti model. The serial interval estimates are only slightly better (figure 2), there are 84% of contacts when considering tracing two days prior to symptoms, compared to what looks like about 80% for the alternative in figure 4 and by the looks of the violin plots from figure 3, quite a bit of overlap if one considers credible intervals. As such, while the analysis is a solid, useful addition to the literature, the authors should provide a better exposition on how it advances scientific insight (the fundamental issues regarding exponential distributions having been identified previously), methodologically (given the thorough analysis by Fraser et al. in 2004) or in terms of impact (given the limited improvement over the Ferretti model).

The reviewer is correct that the problem of independent transmission and symptoms has been noted elsewhere, and that alternative solutions have been proposed (specifically, the pre-print in which the Ferretti model was developed). However, as we state in the revised manuscript (lines 149-152), although the Ferretti pre-print considered both (what we refer to as) the independent transmission and symptoms and Ferretti models, that study did not directly compare predictions generated by those two models. Indeed, to the best of our knowledge, our work is unique in providing explicit comparisons between estimates of key epidemiological quantities and time periods for SARS-CoV-2 infections generated with and without the assumption of independent transmission and symptoms. Since this assumption remains widespread in approaches used to estimate important epidemiological quantities, we believe these comparisons represent a valuable addition to the literature.

As the reviewer points out, our variable infectiousness model provides a modest but definite improvement on the Ferretti model in terms of matching the observed serial intervals. However, since the data contained exposure intervals in addition to symptom onset dates for a significant subset of hosts, the comparisons in Figure 2C only provide a partial picture of the relative goodness of fit between models. In contrast, the AIC values, while not visually intuitive, give a more complete comparison and demonstrate an improved goodness of fit for the variable infectiousness model compared to the Ferretti model. We have emphasised this point in the updated manuscript (lines 156-160).

In addition to the fit to available data, there are clear differences in predictions generated using the variable infectiousness and Ferretti models – for example, our best central estimate for the proportion of presymptomatic transmissions (using the variable infectiousness model) exceeds the upper credible limit predicted by the Ferretti model. This advances scientific insight, since it shows that using a model in which an explicit mechanism links symptoms and infectiousness (e.g. the variable infectiousness model) leads to different estimates of important epidemiological quantities compared to previous approaches.

We thank the reviewer for bringing to our attention the analysis by Fraser et al. (2004), which we cite in our updated submission. In contrast to the comprehensive mathematical treatment of contact tracing (under the assumption of independent transmission and symptoms) in that paper, we provide a simple exposition regarding the consequences of differing model predictions for key aspects of contact tracing and isolation policy.

In our updated submission, we have extended our analysis of isolation and contact tracing in Figure 4 to highlight the impact of our improved fits to SARS-CoV-2 transmission pair data. We have added an entirely new panel in which we consider the impact on transmission of isolating symptomatic hosts (Figure 4A in the updated manuscript). The variable infectiousness model indicates that only 23% of transmissions can be prevented if a symptomatic host is isolated one day after symptom onset, compared to estimates of 38% assuming independent transmission and symptoms, and 28% for the Ferretti model. This limited effectiveness of isolation of symptomatic hosts alone according to our best-fitting model reinforces our conclusion: contact tracing to find presymptomatic infected hosts is very important.

We have also updated the text describing the results shown in Figure 4 to improve the exposition of these results. In the updated manuscript (lines 253-269), we motivate the three key aspects of contact tracing and isolation policy now considered: first, a symptomatic index host must be identified and isolated (Figure 4A); second, contacts up to a specified time before the index host developed symptoms are traced (Figure 4B); third, contacts of the index case are themselves instructed to isolate (Figure 4C). We have also expanded the description of the impact of the results shown in these three panels (lines 271-304).

4. The framing of the paper seems very focused on improving fits to the transmission pair data, however, and I think it would be more impactful to consider the implications of poor estimation of pre-symptomatic transmission and the generation time. I think this shift in focus could also help strengthen the narrative of the paper, which wavers between focusing on model fitting and the importance of implications for contact tracing. Can the authors comment?

We thank the reviewer for this helpful suggestion. We agree that placing more emphasis on the implications of incorrect estimation of factors such as the amount of presymptomatic transmission is helpful to strengthen the narrative of our manuscript. To reflect this, as described in our response to point (3) above, we have expanded both our analyses and discussion of contact tracing and isolation in the updated manuscript. Our new Figure 4A demonstrates how under-estimating the proportion of presymptomatic transmissions leads to the efficacy of isolating symptomatic hosts alone being over-estimated. We also describe the implications of our results for isolation and contact tracing more explicitly in the revised manuscript (lines 271-304).

5. I was a bit lost in the application of the models to the contact tracing example. The definition of the contact elicitation window (lines 142-144), where identification of contacts would occur up to x days prior to contact symptom onset, makes sense theoretically in this model comparison setting, but it is hard to translate these findings to real-world application. Are there any implications that could be useful for informing contact elicitation strategy (e.g., for how many days after time of infection or symptom onset could contact tracing have a measurable benefit in preventing onward transmissions?)

We thank the reviewer for highlighting that the practical implications of our results for contact tracing could be described more clearly. As described in our response to point (3) above, we have expanded both our analyses and discussion of contact tracing in our updated submission.

In the revised manuscript, we have discussed practical considerations for informing contact elicitation strategies (lines 281-290). Predicting how many contacts can be identified using different contact elicitation windows is important for deciding what window to use. Extending the contact elicitation window beyond two days before symptom onset (the current advisory value in the UK and USA) will enable more infected contacts to be identified – for example, we show that if the elicitation window is increased from two to four days, then under the variable infectiousness model the proportion of presymptomatic contacts that can be identified increases from 69% from 93% (lines 281-286).

However, an additional consideration for practical implementation of contact tracing strategies is that a longer contact elicitation window will also lead to more uninfected contacts being instructed to isolate unnecessarily (lines 286-288). This effect is enhanced by the lower infectiousness of index hosts at longer times before symptom onset predicted in Figure 2B (lines 288-290). Furthermore, for contacts that occurred a long time before the index host developed symptoms, the resulting delay before the contact is isolated may limit the effect on onward transmission from the contact, even if they are traced successfully; we discuss an example to illustrate this important point in the updated manuscript (lines 292-304).

6. Lines 147-151: Given that the impact on onward transmission events is so dependent on the contact tracing assumptions, I would recommend stating the assumptions explicitly here, reporting the results in relative terms as compared to a single model, or both.

We have now stated clearly in the revised manuscript that, in the analysis shown in the main text, we have assumed both contact identification and isolation to be 100% effective (lines 253-256). Our results therefore represent the maximum proportion of transmissions that can be prevented for a given delay between onset/infection and isolation, and the maximum proportion of presymptomatic contacts that can be identified for a given elicitation window. In addition, we have presented results in Figure 4—figure supplement 1 in which these assumptions are relaxed (i.e., contact identification and isolation are assumed to have efficacies of below 100%).

7. How different are the variable infectiousness model results from parameter estimates from the original studies that reported the transmission pairs data?

Out of the five original studies which reported the data that we used, only the Ferretti et al. study (Science 368: eabb6936, 2020) estimated the generation time, under the assumption of independent transmission and symptoms. In that study, a mean generation time of 5.0 days (slightly outside our credible interval of 5.1-6.9 days for the independent transmission and symptoms model) and standard deviation of 1.9 days (within our credible interval of 1.8-2.9 days) was obtained assuming independent transmission and symptoms.

As stated in the updated manuscript (lines 149-152), Ferretti et al. also wrote a pre-print in which they fit (what we refer to as) the Ferretti and independent transmission and symptoms models to the same combined dataset that we used, although (as noted in our response to point (3) above) that study did not explicitly compare predictions generated by those two models. To address the reviewer’s comment, we have stated in the revised manuscript (lines 150-151) that the parameter estimates in the Ferretti et al. pre-print lie within the credible intervals that we obtained (these are shown in Supplementary File 1).

8. Can the authors comment on the plausibility of the infectiousness distribution in their new proposed models? While better model fitting certainly provides a measurable improvement to leveraging existing data, I'm not aware of studies that support the discontinuous assumptions about infectiousness made here.

This is a very interesting point. The assumption that the infectiousness of each host changes at most a discrete number of times during infection is common to many widely used compartmental epidemic models. As noted in the revised manuscript (lines 71-72), our mechanistic approach was motivated by compartmental models that incorporate Gamma distributed stage durations and/or changes in infectiousness during infection – in particular, see references (27) and (28) where “SEPIR” compartmental models, in which infectiousness varies between the presymptomatic infectious (P) and symptomatic (I) compartments, were used to model population-level COVID-19 dynamics. The estimated parameter values in our analysis can be used directly in such compartmental epidemiological models.

In our approach, a possible change point in an individual’s infectiousness profile was assumed to coincide exactly with the onset of symptoms. This generated a discontinuous TOST profile, although the generation time distribution (describing the infectiousness profile relative to the time since infection, averaged over a population of hosts) was continuous. As we state in our manuscript (lines 197-199), the estimated change in infectiousness following symptom onset was inferred from the data to be a reduction in infectiousness, which can be attributed to behavioural changes that follow symptom onset.

In reality, behavioural changes may not occur immediately following symptoms, especially if initial symptoms are mild or non-specific. We have therefore outlined in the revised manuscript (lines 389-391) how our mechanistic approach could in principle be extended by adding an additional compartment for early symptomatic hosts. If pre-symptomatic and early symptomatic hosts are assumed to have the same transmission rate (whereas the infectiousness of later symptomatic hosts may be reduced), this would generate a continuous TOST profile. However, we decided not to take this approach here because it would add additional complexity to our mechanistic method, as well as require additional parameters to be estimated (specifically, it would be necessary to estimate a distribution describing the duration of time that hosts spend in the early symptomatic phase). These parameters are unlikely to be identifiable, at least without additional data (see lines 391-393 of the updated manuscript).

9. Assuming alpha means the same thing across the models, why is the 95% credible interval so large for the Feretti model? In general, the model parameters should be more clearly explained for this model. Can the authors comment?

We thank the reviewer for highlighting this potential source of confusion. The parameter previously labelled alpha in the Ferretti model does not have the same meaning as the parameter alpha in our mechanistic approach. We have renamed the parameters in the Ferretti model to alleviate this confusion.

The parameters in the Ferretti model do not have a straightforward epidemiological interpretation (unlike those in the mechanistic approach that we have developed) – we have noted this explicitly in the updated manuscript (line 562-563). We agree that the wide 95% credible interval for the parameter previously labelled alpha in the Ferretti model is interesting, and may suggest identifiability issues – however, detailed analysis of that particular model is outside the scope of this study.